# Fast Parallel Algorithms for Statistical Subset Selection Problems

**Sharon Qian**
Harvard University
sharonqian@g.harvard.edu

**Yaron Singer**
Harvard University
yaron@seas.harvard.edu

## Abstract

In this paper, we propose a new framework for designing fast parallel algorithms for fundamental statistical subset selection tasks that include feature selection and experimental design. Such tasks are known to be *weakly submodular* and are amenable to optimization via the standard greedy algorithm. Despite its desirable approximation guarantees, the greedy algorithm is inherently sequential and in the worst case, its parallel runtime is linear in the size of the data. Recently, there has been a surge of interest in a parallel optimization technique called *adaptive sampling* which produces solutions with desirable approximation guarantees for submodular maximization in exponentially faster parallel runtime. Unfortunately, we show that for general weakly submodular functions such accelerations are impossible. The major contribution in this paper is a novel relaxation of submodularity which we call *differential submodularity*. We first prove that differential submodularity characterizes objectives like feature selection and experimental design. We then design an adaptive sampling algorithm for differentially submodular functions whose parallel runtime is logarithmic in the size of the data and achieves strong approximation guarantees. Through experiments, we show the algorithm's performance is competitive with state-of-the-art methods and obtains dramatic speedups for feature selection and experimental design problems.

## 1 Introduction

In fundamental statistics applications such as regression, classification and maximum likelihood estimation, we are often interested in selecting a subset of elements to optimize an objective function. In a series of recent works, both feature selection (selecting $k$ out of $n$ features) and experimental design (choosing $k$ out of $n$ samples) were shown to be *weakly submodular* [DK11, EKD+18, BBKT17]. The notion of *weak submodularity* was defined by Das and Kempe in [DK11] and quantifies the deviance of an objective function from submodularity. Characterizations of weak submodularity are important as they allow proving guarantees of greedy algorithms in terms of the deviance of the objective function from submodularity. More precisely, for objectives that are $\gamma$-weakly submodular (for $\gamma$ that depends on the objective, see preliminaries Section 2), the greedy algorithm is shown to return a $1 - 1/e^{\gamma}$ approximation to the optimal subset.

**Greedy is sequential and cannot be parallelized.** For large data sets where one wishes to take advantage of parallelization, greedy algorithms are impractical. Greedy algorithms for feature selection such as forward stepwise regression iteratively add the feature with the largest marginal contribution to the objective which requires computing the contribution of each feature in every iteration. Thus, the parallel runtime of the forward stepwise algorithm and greedy algorithms in general, scale linearly with the number of features we want to select. In cases where the computation of the objective function across all elements is expensive or the dataset is large, this can be computationally infeasible.

**Adaptive sampling for fast parallel submodular maximization.** In a recent line of work initiated by [BS18a], adaptive sampling techniques have been used for maximizing submodular functions under varying constraints [BS18b, CQ19b, CQ19a, BBS18, CFK19, ENV19, FMZ19a, BRS19b, EN19, FMZ19b]. Intuitively, instead of growing the solution set element-wise, adaptive sampling adds a large set of elements to the solution at each round which allows the algorithm to be highly parallelizable. In particular, for canonical submodular maximization problems, one can obtain approximation guarantees arbitrarily close to the one obtained by greedy (which is optimal for polynomial time algorithms [NW78]) in *exponentially* faster parallel runtime.

**In general, adaptive sampling fails for weakly submodular functions.** Adaptive sampling techniques add large sets of high valued elements in each round by filtering elements with low marginal contributions. This enables these algorithms to terminate in a small number of rounds. For weak submodularity, this approach renders arbitrarily poor approximations. In Appendix A.1, we use an example of a weakly submodular function from [EDFK17] where adaptive sampling techniques have an arbitrarily poor approximation guarantee. Thus, if we wish to utilize adaptive sampling to parallelize algorithms for applications such as feature selection and experimental design, we need a stronger characterization of these objectives which is amenable to parallelization.

## 1.1 Differential Submodularity

In this paper, we introduce an alternative measure to quantify the deviation from submodularity which we call *differential submodularity*, defined below. We use $f_S(A)$ to denote $f(S \cup A) - f(S)$.

**Definition 1.** *A function $f : 2^N \to \mathbb{R}_+$ is $\alpha$-differentially submodular for $\alpha \in [0, 1]$, if there exist two submodular functions $h, g$ s.t. for any $S, A \subseteq N$, we have that $g_S(A) \geq \alpha \cdot h_S(A)$ and*

$$g_S(A) \leq f_S(A) \leq h_S(A)$$

A 1-differentially submodular function is submodular and a 0-differentially submodular function can be arbitrarily far from submodularity. In Figure 1, we show a depiction of differential submodularity (blue lines) calculated from the feature selection objective by fixing an element $a$ and randomly sampling sets $S$ of size 100 to compute the marginal contribution $f_S(a)$ on a real dataset. For a differentially submodular function (blue lines), the property of decreasing marginal contributions does not hold but can be bounded by two submodular functions (red) with such property.

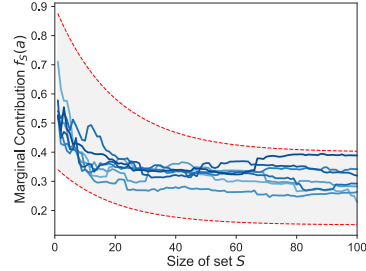

As we prove in this paper, applications such as feature selection for regression and classification as well as experimental design are all $\gamma^2$-differentially submodular, where $\gamma$ corresponds to their weak submodularity ratios [EKD+18, BBKT17]. The power of this characterization is that it allows for parallelization with strong approximation guarantees. We do this by designing an adaptive sampling algorithm that leverages the differential submodularity structure and has bounded approximation guarantees in terms of the differential submodularity ratios.

Figure 1: Marginal contribution of differentially submodular function.

## 1.2 Main results

Our main result is that for objectives such as feature selection for regression and classification and Bayesian A-optimality experimental design which are all $\gamma$-weakly submodular, there is an approximation guarantee arbitrarily close to $1 - 1/e^{\gamma^4}$ for maximization under cardinality constraints in $\mathcal{O}(\log n)$ adaptive rounds (see adaptivity definition in Section 2). Thus, while the approximation is inferior to the $1 - 1/e^{\gamma}$ obtained by greedy, our algorithm has exponentially fewer rounds. Importantly, using experiments we show that empirically it has comparable terminal values to the greedy algorithm, greatly outperforms its theoretical lower bound, and obtains the result with two to eight-fold speedups. We achieve our result by proving these objectives are $\alpha$-differentially submodular and designing an adaptive sampling algorithm that gives a $1 - 1/e^{\alpha^2}$ approximation for maximizing any $\alpha$-differentially submodular function under a cardinality constraint.

**Conceptual overview.** For the past decade, fundamental problems in machine learning have been analyzed through relaxed notions of submodularity (See details on different relaxations of submodu-

larity and relationship to differential submodularity in Appendix B). Our main conceptual contribution is the framework of differential submodularity which is purposefully designed to enable fast parallelization techniques that previously-studied relaxations of submodularity do not. Specifically, although stronger than weak submodularity, we can prove direct relationships between objectives' weak submodularity ratios and their differential submodularity ratios which allows getting strong approximations and exponentially faster parallel runtime. We note that differential submodularity is also applicable to more recent parallel optimization techniques such as adaptive sequencing [BRS19b].

**Technical overview.** From a purely technical perspective, there are two major challenges addressed in this work. The first pertains to the characterization of the objectives in terms of differential submodularity and the second is the design of an adaptive sampling algorithm for differentially submodular functions. Previous adaptive sampling algorithms are purposefully designed for submodular functions and cannot be applied when the objective function is not submodular (example in Appendix A.2). In these cases, the marginal contribution of individual elements is not necessarily subadditive to the marginal contribution of the set of elements combined. Thus, the standard analysis of adaptive sampling, where we attempt to add large sets of elements to the solution set by assessing the value of individual elements, does not hold. By leveraging the fact that marginal contributions of differentially submodular functions can be bounded by marginal contributions of submodular functions, we can approximate the marginal contribution of a set by assessing the marginal contribution of its elements. This framework allows us to leverage parallelizable algorithms to show a stronger approximation guarantee in exponentially fewer rounds.

**Paper organization.** We first introduce preliminary definitions in Section 2 followed by introducing our main framework of differential submodularity and its reduction to feature selection and experimental design objectives in Section 3. We then introduce an algorithm for selection problems using adaptive sampling in Section 4 and conclude with experiments in Section 5. Due to space constraints, most proofs of the analysis are deferred to the Appendix.

## 2   Preliminaries

For a positive integer $n$, we use $[n]$ to denote the set $\{1, 2, \ldots, n\}$. Boldface lower and upper case letters denote vectors and matrices respectively: $\mathbf{a}, \mathbf{x}, \mathbf{y}$ represent vectors and $\mathbf{A}, \mathbf{X}, \mathbf{Y}$ represent matrices. Unbolded lower and upper case letters present elements and sets respectively: $a, x, y$ represent elements and $A, X, Y$ represent sets. For a matrix $\mathbf{X} \in \mathbb{R}^{d \times n}$ and $S \subseteq [n]$, we denote submatrices by column indices by $\mathbf{X}_S$. For vectors, we use $\mathbf{x}_S$ to denote supports $\text{supp}(\mathbf{x}) \subseteq S$. To connect the discrete function $f(S)$ to a continuous function, we let $f(S) = \ell(\mathbf{w}^{(S)})$, where $\mathbf{w}^{(S)}$ denotes the $\mathbf{w}$ that maximizes $\ell(\cdot)$ subject to $\text{supp}(\mathbf{w}) \subseteq S$.

**Submodularity and weak submodularity.** A function $f : 2^N \to \mathbb{R}_+$ is submodular if $f_S(a) \geq f_T(a)$ for all $a \in N \setminus T$ and $S \subseteq T \subseteq N$. It is *monotone* if $f(S) \leq f(T)$ for all $S \subseteq T$. We assume that $f$ is normalized and non-negative, i.e., $0 \leq f(S) \leq 1$ for all $S \subseteq N$, and monotone. The concept of *weak submodularity* is a relaxation of submodularity, defined via the *submodularity ratio*:

**Definition 2.** *[DK11] The **submodularity ratio** of $f : 2^N \to \mathbb{R}_+$ is defined as, for all $A \subseteq N$,*

$$\gamma_k = \min_{A \subseteq N, S:|A| \leq k} \frac{\sum_{a \in A} f_S(a)}{f_S(A)}.$$

Functions with submodularity ratios $\gamma = \min_k \gamma_k < 1$ are $\gamma$-*weakly submodular*.

**Adaptivity.** The *adaptivity* of algorithms refers to the number of sequential rounds of queries it makes when polynomially-many queries can be executed in parallel in each round.

**Definition 3.** *For a function $f$, an algorithm is $r$-adaptive if every query $f(S)$ given a set $S$ occurs at a round $i \in [r]$ such that $S$ is independent of the values $f(S')$ of all other queries at round $i$.*

Adaptivity is an information theoretic measure of parallel-runtime that can be translated to standard parallel computation frameworks such as PRAM (See Appendix C). Therefore, like all previous work on adaptivity on submodular maximization, we are interested in algorithms that have low adaptivity since they are parallelizable and scalable for large datasets [BRS19a, BS18b, CQ19b, CQ19a, BBS18, CFK19, ENV19, FMZ19a, BRS19b, EN19, FMZ19b].

# 3 Feature Selection and A-Optimal Design are Differentially Submodular

We begin by characterizing differential submodularity in terms of *restricted strong concavity* and *restricted smoothness* defined as follows.

**Definition 4.** *[EKD+18] Let $\Omega$ be a subset of $\mathbb{R}^n \times \mathbb{R}^n$ and $\ell : \mathbb{R}^n \to \mathbb{R}$ be a continuously differentiable function. A function $\ell$ is **restricted strong concave (RSC)** with parameter $m_\Omega$ and **restricted smooth (RSM)** with parameter $M_\Omega$ if, for all $(\mathbf{y}, \mathbf{x}) \in \Omega$,*

$$-\frac{m_\Omega}{2}\|\mathbf{y} - \mathbf{x}\|_2^2 \quad \geq \quad \ell(\mathbf{y}) - \ell(\mathbf{x}) - \langle \nabla\ell(\mathbf{x}), \mathbf{y} - \mathbf{x}\rangle \geq -\frac{M_\Omega}{2}\|\mathbf{y} - \mathbf{x}\|_2^2$$

Before connecting our notion of differential submodularity to RSC/RSM properties, we first define concavity and smoothness parameters on subsets of $\Omega$. If $\Omega' \subseteq \Omega$, then $M_{\Omega'} \leq M_\Omega$ and $m_{\Omega'} \geq m_\Omega$.

**Definition 5.** *We define the domain of $s$-sparse vectors as $\Omega_s = \{(\mathbf{x}, \mathbf{y}) : \|\mathbf{x}\|_0 \leq s, \|\mathbf{y}\|_0 \leq s, \|\mathbf{x} - \mathbf{y}\|_0 \leq s\}$. If $t \geq s$, $M_s \leq M_t$ and $m_s \geq m_t$.*

**Theorem 6.** *Suppose $\ell(\cdot)$ is RSC/RSM on $s$-sparse subdomains $\Omega_s$ with parameters $m_s, M_s$ for $s \leq 2k$. Then, for $t = |S|+k, s = |S|+1$, the objective $f(S) = \ell(\mathbf{w}^{(S)})$ is differentially submodular s.t. for $S, A \subseteq N$, $|A| \leq k$, $\frac{m_s}{M_t}\tilde{f}_S(A) \leq f_S(A) \leq \frac{M_s}{m_t}\tilde{f}_S(A)$, where $\tilde{f}_S(A) = \sum_{a \in A} f_S(a)$.*

*Proof.* We first prove the lower bound of the inequality. We define $\mathbf{x}_{(S \cup A)} = \frac{1}{M_t}\nabla\ell(\mathbf{w}^{(S)})_A + \mathbf{w}^{(S)}$ and use the strong concavity of $\ell(\cdot)$ to lower bound $f_S(A)$:

$$f_S(A) \geq \ell(\mathbf{x}_{(S \cup A)}) - \ell(\mathbf{w}^{(S)}) \quad \geq \quad \langle \nabla\ell(\mathbf{w}^{(S)}), \mathbf{x}_{(S \cup A)} - \mathbf{w}^{(S)}\rangle - \frac{M_t}{2}\|\mathbf{x}_{(S \cup A)} - \mathbf{w}^{(S)}\|_2^2$$

$$\geq \quad \frac{1}{2M_t}\|\nabla\ell(\mathbf{w}^{(S)})_A\|_2^2 \tag{1}$$

where the first inequality follows from the optimality of $\ell(\mathbf{w}^{(S \cup A)})$ for vectors with support $S \cup A$ and the last inequality is by the definition of $\mathbf{x}_{(S \cup A)}$.

We also can use smoothness of $\ell(\cdot)$ to upper bound the marginal contribution of each element in $A$ to $S$, $f_S(a)$. We define $\mathbf{x}_{(S \cup a)} = \frac{1}{m_s}\nabla\ell(\mathbf{w}^{(S)})_a + \mathbf{w}^{(S)}$. For $a \in A$,

$$f_S(a) = \ell(\mathbf{w}^{(S \cup a)}) - \ell(\mathbf{w}^{(S)}) \quad \leq \quad \langle \nabla\ell(\mathbf{w}^{(S)}), \mathbf{x}_{(S \cup a)} - \mathbf{w}^{(S)}\rangle - \frac{m_s}{2}\|\mathbf{x}_{(S \cup a)} - \mathbf{w}^{(S)}\|_2^2$$

$$\leq \quad \frac{1}{2m_s}\|\nabla\ell(\mathbf{w}^{(S)})_a\|_2^2 \tag{2}$$

where the last inequality follows from the definition of $\mathbf{x}_{(S \cup a)}$. Summing across all $a \in A$, we get

$$\sum_{a \in A} f_S(a) \quad \leq \quad \sum_{a \in A}\frac{1}{2m_s}\|\nabla\ell(\mathbf{w}^{(S)})_a\|_2^2 = \frac{1}{2m_s}\|\nabla\ell(\mathbf{w}^{(S)})_A\|_2^2 \tag{3}$$

By combining (1) and (3), we can get the desired lower bound of $f_S(A)$. To get the upper bound on the marginals, we can use the lower bound of submodularity ratio $\gamma_{S,k}$ of $f$ from Elenberg et al. [EKD+18], which is no less than $\frac{m_t}{M_s}$. Then, by letting $\tilde{f}_S(A) = \sum_{a \in A} f_S(a)$, we can complete the proof and show that the marginals can be bounded. $\square$

We can further generalize the previous lemma to all sets $S, A \subseteq N$, by using the general RSC/RSM parameters $m, M$ associated with $\Omega_n$, where $n \geq t, s$. From Definition 5, since $\Omega_s \subseteq \Omega_t \subseteq \Omega_n$, $M_s \leq M_t \leq M$ and $m_s \geq m_t \geq m$. Thus, we can weaken the bounds from Lemma 6 to get $\frac{m}{M}\tilde{f}_S(A) \leq f_S(A) \leq \frac{M}{m}\tilde{f}_S(A)$ which is a $\gamma^2$-differentially submodular function for $\gamma = \frac{m}{M}$.

## 3.1 Differential submodularity bounds for statistical subset selection problems

We now connect differential submodularity to feature selection and experimental design objectives. We also show that even when adding diversity-promoting terms $d(S)$ as in [DDK12] the functions remain differentially submodular. Due to space limitations, proofs are deferred to Appendix E.

**Feature selection for regression.** For a response variable $\mathbf{y} \in \mathbb{R}^d$ and feature matrix $\mathbf{X} \in \mathbb{R}^{d \times n}$, the objective is the maximization of the $\ell_2$-utility function that represents the variance reduction of $\mathbf{y}$ given the feature set $S$:

$$\ell_{\texttt{reg}}(\mathbf{y}, \mathbf{w}^{(S)}) = \|\mathbf{y}\|_2^2 - \|\mathbf{y} - \mathbf{X}_S \mathbf{w}\|_2^2$$

We can bound the marginals by eigenvalues of the feature covariance matrix. We denote the minimum and maximum eigenvalues of the $k$-sparse feature covariance matrix by $\lambda_{min}(k)$ and $\lambda_{max}(k)$.

**Corollary 7.** *Let $\gamma = \frac{\lambda_{min}(2k)}{\lambda_{max}(2k)}$ and $d : 2^N \to \mathbb{R}_+$ be a submodular diversity function. Then $f(S) = \ell_{reg}(\mathbf{w}^{(S)})$ and $f_{div}(S) = \ell_{reg}(\mathbf{w}^{(S)}) + d(S)$ are $\gamma^2$-differentially submodular.*

We note that [DK11] use a different objective function to measure the goodness of fit $R^2$. In Appendix F, we show an analogous bound for the objective used in [DK11]. Our lower bound is consistent with the result in Lemma 2.4 from Das and Kempe [DK11].

**Feature selection for classification.** For classification, we wish to select the best $k$ columns from $\mathbf{X} \in \mathbb{R}^{d \times n}$ to predict a categorical variable $\mathbf{y} \in \mathbb{R}^d$. We use the following log-likelihood objective in logistic regression to select features. For a categorical variable $\mathbf{y} \in \mathbb{R}^d$, the objective in selecting the elements to form a solution set is the maximization of the log-likelihood function for a given $S$:

$$\ell_{\texttt{class}}(\mathbf{y}, \mathbf{w}^{(S)}) = \sum_{i=1}^{d} y_i (\mathbf{X}_S \mathbf{w}) - \log(1 + e^{\mathbf{X}_S \mathbf{w}})$$

We denote $m$ and $M$ to be the RSC/RSM parameters on the feature matrix $\mathbf{X}$. For $\gamma = \frac{m}{M}$ [EKD$^+$18] show that the feature selection objective for classification is $\gamma$-weakly submodular.

**Corollary 8.** *Let $\gamma = \frac{m}{M}$ and $d : 2^N \to \mathbb{R}_+$ be a submodular diversity function. Then $f(S) = \ell_{class}(\mathbf{w}^{(S)})$ and $f_{div}(S) = \ell_{class}(\mathbf{w}^{(S)}) + d(S)$ are $\gamma^2$-differentially submodular.*

**Bayesian A-optimality for experimental design.** In experimental design, we wish to select the set of experimental samples $\mathbf{x}_i$ from $\mathbf{X} \in \mathbb{R}^{d \times n}$ to maximally reduce variance in the parameter posterior distribution. We now show that the objective for selecting diverse experiments using Bayesian A-optimality criterion is differentially submodular. We denote $\mathbf{\Lambda} = \beta^2 \mathbf{I}$ as the prior that takes the form of an isotropic Gaussian and $\sigma^2$ as variance (See Appendix D for more details).

**Corollary 9.** *Let $\gamma = \frac{\beta^2}{\|\mathbf{X}\|^2 (\beta^2 + \sigma^{-2} \|\mathbf{X}\|^2)}$ and $d : 2^N \to \mathbb{R}_+$ be a submodular diversity function, then the objectives of Bayesian A-optimality defined by $f_{A\text{-}opt}(S) = \text{Tr}(\mathbf{\Lambda}^{-1}) - \text{Tr}((\mathbf{\Lambda} + \sigma^{-2}\mathbf{X}_S \mathbf{X}_S^T)^{-1})$ and the diverse analog defined by $f_{A\text{-}div}(S) = f_{A\text{-}opt}(S) + d(S)$ are $\gamma^2$-differentially submodular.*

## 4 The Algorithm

We now present the DASH (DIFFERENTIALLY-ADAPTIVE-SHAMPLING) algorithm for maximizing differentially submodular objectives with logarithmic adaptivity. Similar to recent works on low adaptivity algorithms [BRS19a, BS18b, CQ19b, CQ19a, BBS18, CFK19, ENV19, FMZ19a, BRS19b, EN19], this algorithm is a variant of the adaptive sampling technique introduced in [BS18a]. The adaptive sampling algorithm for submodular functions, where $\alpha = 1$, is not guaranteed to terminate for non-submodular functions (See Appendix A.2). Thus, we design a variant to specifically address differential submodularity to parallelize the maximization of non-submodular objectives.

**Algorithm overview.** At each round, the DASH algorithm selects good elements determined by their individual marginal contributions and attempts to add a set of $k/r$ elements to the solution set $S$. The decision to label elements as "good" or "bad" depends on the threshold $t$ which quantifies the distance between the elements that have been selected and OPT. This elimination step takes place in the `while` loop and effectively filters out elements with low marginal contributions. The algorithm terminates when $k$ elements have been selected or when the value of $f(S)$ is sufficiently close to OPT.

The algorithm presented is an idealized version because we cannot exactly calculate expectations, and OPT and differential submodularity parameter $\alpha$ are unknown. We can estimate the expectations by increasing sampling of the oracle and we can guess OPT and $\alpha$ through parallelizing multiple guesses (See Appendix G for more details).

---
**Algorithm 1** DASH $(N, r, \alpha)$
---
1: **Input** Ground set $N$, number of outer-iterations $r$, differential submodularity parameter $\alpha$
2: $S \leftarrow \emptyset, X \leftarrow N$
3: **for** $r$ iterations **do**
4:      $t := (1 - \epsilon)(f(O) - f(S))$
5:      **while** $\mathbb{E}_{R \sim \mathcal{U}(X)}[f_S(R)] < \alpha^2 \frac{t}{r}$ **do**
6:          $X \leftarrow X \backslash \{a : \mathbb{E}_{R \sim \mathcal{U}(X)}[f_{S \cup (R \backslash \{a\})}(a)] < \alpha(1 + \frac{\epsilon}{2})t/k\}$
7:      **end while**
8:      $S \leftarrow S \cup R$ where $R \sim \mathcal{U}(X)$
9: **end for**
10: **return** $S$
---

**Algorithm analysis.** We now outline the proof sketch of the approximation guarantee of $f(S)$ using DASH. In our analysis, we denote the optimal solution as $\mathtt{OPT} = f(O)$ where $O = \text{argmax}_{|S| \leq k} f(S)$ and $k$ is a cardinality constraint parameter. Proof details can be found in Appendix H.

**Theorem 10.** *Let $f$ be a monotone, $\alpha$-differentially submodular function where $\alpha \in [0, 1]$, then, for any $\epsilon > 0$, DASH is a $\log_{1+\epsilon/2}(n)$ adaptive algorithm that obtains the following approximation for the set $S$ that is returned by the algorithm*

$$f(S) \geq (1 - 1/e^{\alpha^2} - \epsilon)f(O).$$

The key adaptation for $\alpha$-differential submodular functions appears in the thresholds of the algorithm, one to filter out elements and another to lower bound the marginal contribution of the set added in each round. The additional $\alpha$ factor in the `while` condition compared to the single element marginal contribution threshold is a result of differential submodularity properties and guarantees termination.

To prove the theorem, we lower bound the marginal contribution of selected elements $X_\rho$ at each iteration $\rho$: $f_S(X_\rho) \geq \frac{\alpha^2}{r}(1 - \epsilon)(f(O) - f(S))$ (Lemma 19 in Appendix H.1).

We can show that the algorithm terminates in $\log_{1+\epsilon/2}(n)$ rounds (Lemma 21 in Appendix H.1). Then, using the lower bound of the marginal contribution of a set at each round $f_S(X_\rho)$ in conjunction with an inductive proof, we get the desired result.

We have seen in Corollary 7, 8 and 9 that the feature selection and Bayesian experimental design problems are differentially submodular. Thus, we can apply DASH to these problems to obtain the $f(S) \geq (1 - 1/e^{\alpha^2} - \epsilon)f(O)$ guarantee from Theorem 10.

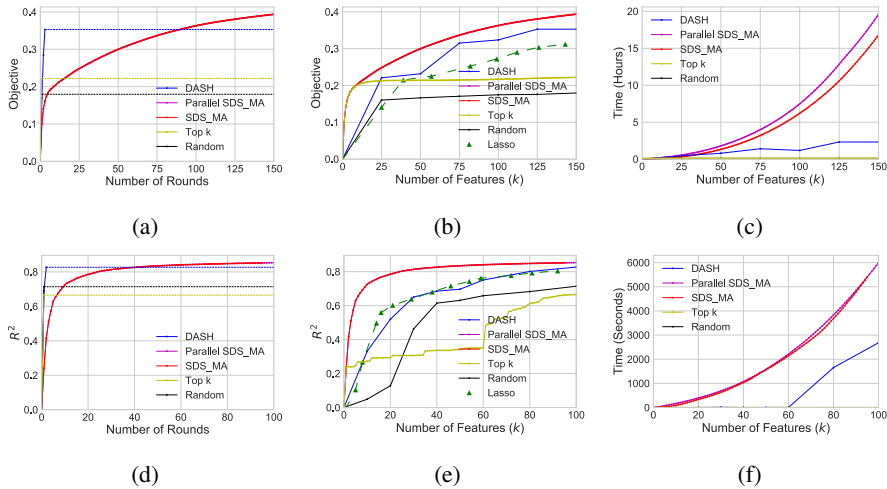

Figure 2: Linear regression feature selection results comparing DASH (blue) to baselines on synthetic (top row) and clinical datasets (bottom row). Dashed line represents LASSO extrapolated across $\lambda$.

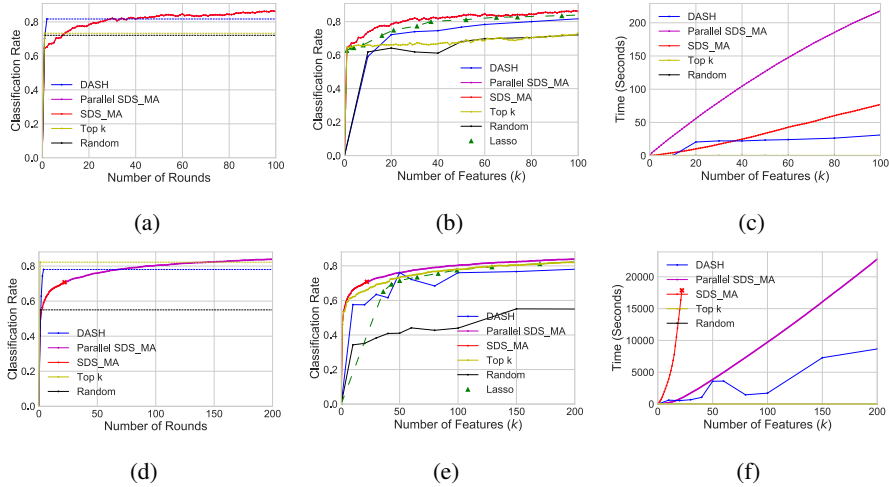

Figure 3: Logistic regression feature selection results comparing DASH (blue) to baselines on synthetic (top row) and gene datasets (bottom row). The X denotes manual termination of the algorithm due to running time constraints. Dashed line represents approximation for LASSO extrapolated across $\lambda$.

## 5 Experiments

To empirically evaluate the performance of DASH, we conducted several experiments on feature selection and Bayesian experimental design. While the $1 - 1/e^{\gamma^4}$ approximation guarantee of DASH is weaker than the $1 - 1/e^\gamma$ of the greedy algorithm (SDS$_{MA}$), we observe that DASH performs comparably to SDS$_{MA}$ and outperforms other benchmarks. Most importantly, in all experiments, DASH achieves a two to eight-fold speedup of **parallelized** greedy implementations, even for moderate values of $k$. This shows the incredible potential of other parallelizable algorithms, such as adaptive sampling and adaptive sequencing, under the differential submodularity framework.

**Datasets.** We conducted experiments for linear and logistic regression using the $\ell_{\texttt{reg}}$ and $\ell_{\texttt{class}}$ objectives, and Bayesian experimental design using $f_{\texttt{A-opt}}$. We generated the synthetic feature space from a multivariate normal distribution. To generate the response variable $\mathbf{y}$, we sample coefficients uniformly (D1) and map to probabilities for classification (D3) and attempt to select important features and samples. We also select features on a clinical dataset $n = 385$ (D2) and classify location of cancer in a biological dataset $n = 2500$ (D4). We use D1, D2 for linear regression and Bayesian experimental design, and D3, D4 for logistic regression experiments. (See Appendix I.2 for details.)

**Benchmarks.** We compared DASH to RANDOM (selecting $k$ elements randomly in one round), TOP-$k$ (selecting $k$ elements of largest marginal contribution), SDS$_{MA}$ [KC10] and Parallel SDS$_{MA}$, and LASSO, a popular algorithm for regression with an $\ell_1$ regularization term. (See Appendix I.3.)

**Experimental Setup.** We run DASH and baselines for different $k$ for two sets of experiments.

- **Accuracy vs. rounds.** In this set of experiments, for each dataset we fixed one value of $k$ ($k = 150$ for D1, $k = 100$ for D2, D3 and $k = 200$ for D4) and ran algorithms to compare accuracy of the solution ($R^2$ for linear regression, classification rate for logistic regression and Bayesian A-optimality for experimental design) as a function of the number of parallel rounds. The results are plotted in Figures 2a, 2d, Figures 3a, 3d and Figures 4a, 4d;

- **Accuracy and time vs. features.** In these experiments, we ran the same benchmarks for varying values of $k$ (in D1 the maximum is $k = 150$, D2, D3 the maximum is $k = 100$ and in D4 the maximum is $k = 200$) and measure both accuracy (Figures 2b, 2e, 3b, 3e, 4b, 4e) and time (Figures 2c, 2f, 3c, 3f, 4c, 4f). When measuring accuracy, we also ran LASSO by manually varying the regularization parameter $\lambda$ to select approximately $k$ features. Since each $k$ represents a different run of the algorithm, the output (accuracy or time) is not necessarily monotonic with respect to $k$.

We implemented DASH with 5 samples at every round. Even with this small number of samples, the terminal value outperforms greedy throughout all experiments. The advantage of using fewer samples is that it allows parallelizing over fewer cores. In general, given more cores one can reduce the variance in estimating marginal contributions which improves the performance of the algorithm.

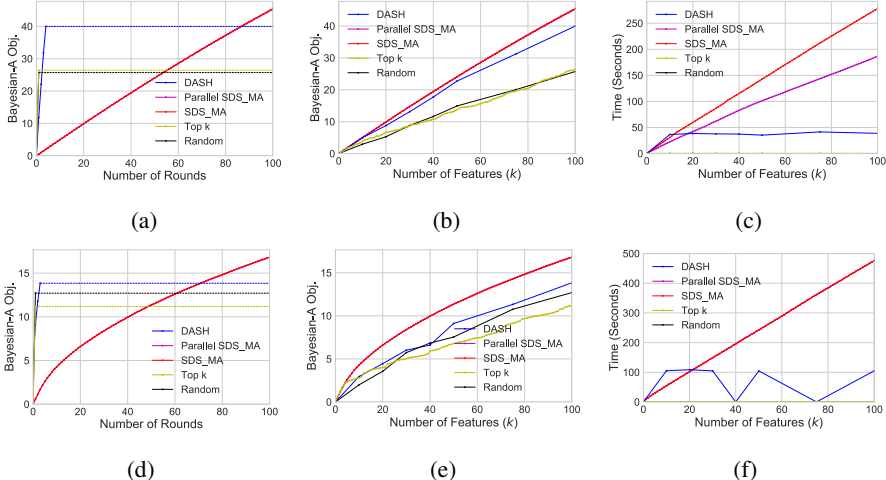

Figure 4: Bayesian experimental design results comparing DASH (blue) to baselines on synthetic (top row) and clinical datasets (bottom row).

**Results on general performance.** We first analyze the performance of DASH. For all applications, Figures 2a, 2d, 3a, 3d, 4a and 4d show that the final objective value of DASH is comparable to $SDS_{MA}$, outperforms TOP-$k$ and RANDOM, and is able to achieve the solution in much fewer rounds. In Figures 2b, 2e, 3b, 3e, 4b and 4e, we show DASH can be very practical in finding a comparable solution set to $SDS_{MA}$ especially for larger values of $k$. In the synthetic linear regression experiment, DASH significantly outperforms LASSO and has comparable performance in other experiments. While DASH outperforms the simple baseline of RANDOM, we note that the performance of RANDOM varies widely depending on properties of the dataset. In cases where a small number of features can give high accuracy, RANDOM can perform well by randomly selecting well-performing features when $k$ is large (Figure 2e). However, in more interesting cases where the value does not immediately saturate, both DASH and $SDS_{MA}$ significantly outperform RANDOM (Figure 2b, 4b).

We can also see in Figures 2c, 2f, 3c, 3f, 4c and 4f that DASH is computationally efficient compared to the other baselines. In some cases, for smaller values of $k$, $SDS_{MA}$ is faster (Figure 3c). This is mainly due to the sampling done by DASH to estimate the marginals, which can be computationally intensive. However, in most experiments, DASH terminates more quickly even for small values of $k$. For larger values, DASH shows a two to eight-fold speedup compared to the fastest baseline.

**Effect of oracle queries.** Across our experiments, the cost for oracle queries vary widely. When the calculation of the marginal contribution is computationally cheap, parallelization of $SDS_{MA}$ has a longer running time than its sequential analog due to the cost of merging parallelized results (Figures 2c, 3c). However, in the logistic regression gene selection experiment, calculating the marginal contribution of an element to the solution set can span more than 1 minute. In this setting, using sequential $SDS_{MA}$ to select 100 elements would take several days for the algorithm to terminate (Figure 3f). Parallelization of $SDS_{MA}$ drastically improves the algorithm running time, but DASH is still much faster and can find a comparable solution set in under half the time of parallelized $SDS_{MA}$.

In both cases of cheap and computationally intensive oracle queries, DASH terminates more quickly than the sequential and parallelized version of $SDS_{MA}$ for larger values of $k$. This can be seen in Figures 2c, 3c and 4c where calculation of marginal contribution on synthetic data is fast and in Figures 2f, 3f and 4f where oracle queries on larger datasets are much slower. This shows the incredible potential of using DASH across a wide array of different applications to drastically cut down on computation time in selecting a large number elements across different objective functions. Given access to more processors, we expect even a larger increase in speedup for DASH.

## Acknowledgements

The authors would like to thank Eric Balkanski for helpful discussions. This research was supported by a Smith Family Graduate Science and Engineering Fellowship, NSF grant CAREER CCF 1452961, NSF CCF 1301976, BSF grant 2014389, NSF USICCS proposal 1540428, a Google Research award, and a Facebook research award.

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
