[Supplementary Material]

# A Motivational Examples

## A.1 ADAPTIVE-SAMPLING does not work for weakly submodular functions

To demonstrate why adding sets at each iteration can perform badly compared to adding single elements, we construct a weakly submodular function where greedy can achieve the optimal value and the performance of adding sets of elements to the solution set can be poor. The construction is a slight variant of the one in [EDFK17].

We have a ground set consisting of two types of elements, $\mathcal{N} = \{U, V\}$, where $U = \{u_i\}_{i=1}^k$ and $V = \{v_i\}_{i=1}^k$. For every subset $S \subseteq \mathcal{N}$, $u(S) = |S \cap U|$ and $v(S) = |S \cap V|$. Now, we define the following set function

$$f(S) = \min\{2 \cdot u(S) + 1, 2 \cdot v(S)\}, \quad \forall S \subseteq N.$$

For cardinality constraint $k$, we can see that the optimal solution is $k$.

**Lemma 11.** *$f$ is nonnegative, monotone and 0.5-weakly submodular [EDFK17].*

For simplicity, assume the number of rounds $r = 1$. We now show why ADAPTIVE SAMPLING performs poorly. In the first step, ADAPTIVE SAMPLING will filter out elements with low marginal contributions. Since $f(u_i) = 0$ and $f(v_i) = 1$ for all $i$, by standard concentration bounds, elements of $U$ will be filtered out and only elements in $V$ will remain. Now, the algorithm attempts to add a set of $k$ elements into the solution set. Since all subsets of $V$ have a value of 1, the algorithm can only achieve a value of 1 even when the optimal value is $k$. As $k$ increases, this algorithm performs arbitrarily poorly.

## A.2 Existing adaptive algorithms fail for differentially submodular functions

ADAPTIVE-SAMPLING [BRS19a] for submodular functions does not guarantee termination for differentially submodular functions. The filtering step that removes elements with low individual marginal contribution does not guarantee the marginal contribution of the set of "good" elements is larger than the threshold value as in the submodular case. For differentially submodular functions, the algorithm may result in an infinite `while` loop, where "bad" elements are filtered out, but no combination of remaining elements adds sufficient value to the solution set. We show two examples.

We use the construction defined in the previous section $f(S) = \min\{2 \cdot u(S) + 1, 2 \cdot v(S)\}$. While $f(S)$ is weakly submodular, we note that it is not differentially submodular. Consider the case where $S = \{u_1\}$ and $A = \{v_i\}_{i=1}^n$, then $\sum_{a \in A} f_S(a) = n$, but $f_S(A) = 1$. However, we can show a modified function is differentially submodular on small set sizes, which is sufficient for our example. Let $f'(S) = f(S)$ where $|S| \leq 2$, then $f'$ is 0.25-differentially submodular. This construction demonstrates a simple case of how adaptive sampling on submodular functions fails for differentially submodular functions.

**Lemma 12.** *$f'$ is 0.25-differentially submodular.*

*Proof.* We can use the lower bound from weak submodularity of $f$ from Lemma 11, which holds for $f'$:

$$\sum_{a \in A} f'_S(a) \geq 0.5 \cdot f'_S(A)$$

With our modification, we can now lower bound by marginal contributions:

$$f'_S(A) \geq \frac{1}{|A \setminus S|} \cdot \sum_{a \in A} f'_S(a) \geq 0.5 \cdot \sum_{a \in A} f'_S(a)$$

which shows that $f'$ is 0.25-differentially submodular. $\square$

We now show that ADAPTIVE-SAMPLING does not guarantee termination for differentially submodular functions.

For simplicity, let $\epsilon = 0$. We wish to select $k$ elements to achieve the optimal solution of $k$ by adding 2 elements at a time to the solution set using ADAPTIVE-SAMPLING on $f'(S)$. We note that DASH reduces to ADAPTIVE-SAMPLING when $\alpha = 1$.

To survive the filtering step, each element must have a marginal contribution of 1. Since $f'(v_i) = 1$ and $f'(u_i) = 0$ for all $i$, only elements in $V$ are labeled as "good" by the algorithm. The elements in $U$ are filtered out. Then the algorithm attempts to add 2 elements from $V$ into the solution set and expects that the marginal contribution of the set has value 2 for termination (for $\alpha = 1$). However, this is not the case, as $f'(v_i \cup v_j) = 1$ and ADAPTIVE-SAMPLING enters an infinite `while` loop by failing to find a set with large enough marginal contribution.

However, DASH will terminate. By adding a factor of $\alpha^2$ to lower the threshold, DASH accepts the set of 2 elements in $V$ and successfully adds these 2 elements into the solution set. The algorithm leverages the fact that differential submodularity both lower bounds the elements that are added into the set and upper bounds the values of elements that are filtered out.

In another more concrete example, we show that after individual elements are filtered out, there is no set of elements that will pass the ADAPTIVE-SAMPLING threshold to be added into the solution set. This results in an infinite `while` loop.

Consider the following variables in the context of the $R^2$, goodness-of-fit objective (See Appendix F for more details):

$$
\begin{aligned}
\mathbf{y} &= \begin{bmatrix} 1 & 0 & 0 & 0 \end{bmatrix}^\top \\
\mathbf{x}_1 &= \begin{bmatrix} 0 & 1 & 0 & 0 \end{bmatrix}^\top \\
\mathbf{x}_2 &= \begin{bmatrix} 0 & 0 & 1 & 0 \end{bmatrix}^\top \\
\mathbf{x}_3 &= \begin{bmatrix} 0 & 0 & 0 & 1 \end{bmatrix}^\top \\
\mathbf{x}_4 &= \begin{bmatrix} \sqrt{\frac{1}{2}} & \sqrt{\frac{1}{2}} & 0 & 0 \end{bmatrix}^\top \\
\mathbf{x}_5 &= \begin{bmatrix} \sqrt{\frac{1}{2}} & 0 & \sqrt{\frac{1}{2}} & 0 \end{bmatrix}^\top \\
\mathbf{x}_6 &= \begin{bmatrix} \sqrt{\frac{1}{2}} & 0 & 0 & \sqrt{\frac{1}{2}} \end{bmatrix}^\top
\end{aligned}
$$

We wish to choose two features $\mathbf{x}_i$ that best estimate $\mathbf{y}$ (and maximize $R^2$). We can see that the optimal solution of $R^2 = 1$ is achieved by three different 2-subsets: $(\mathbf{x}_1, \mathbf{x}_4), (\mathbf{x}_2, \mathbf{x}_5), (\mathbf{x}_3, \mathbf{x}_6)$. For $\mathbf{x}_1, \mathbf{x}_2, \mathbf{x}_3$, the marginal contribution is $R^2 = 0$. For $\mathbf{x}_4, \mathbf{x}_5, \mathbf{x}_6$, $R^2 = \frac{1}{2}$.

For simplicity, let $\epsilon = 0$, $r = 1$ and $f(O) = 1$. ADAPTIVE-SAMPLING will first filter out $\mathbf{x}_1, \mathbf{x}_2, \mathbf{x}_3$ because the marginal contribution is less than $\frac{1}{2}$. Then it will attempt to select 2 elements from $\mathbf{x}_4, \mathbf{x}_5, \mathbf{x}_6$ to comprise the solution set. The while loop will only terminate once it finds a 2-subset where the marginal contribution is larger or equal to 1. However, due to the non-submodular properties of the objective, even though the bad elements were filtered out, the marginal contribution of any 2-subset from $\mathbf{x}_4, \mathbf{x}_5, \mathbf{x}_6$ does not achieve the necessary threshold value. The $R^2$ of any 2-subset from $\mathbf{x}_4, \mathbf{x}_5, \mathbf{x}_6$ is $\frac{2}{3}$. As an example, let us calculate the marginal contribution of $\mathbf{x}_4$ and $\mathbf{x}_5$.

$$
\begin{aligned}
R_{4,5}^2 &= (\mathbf{y}^\top \mathbf{X}_{4,5})(\mathbf{X}_{4,5}^\top \mathbf{X}_{4,5})^{-1}(\mathbf{X}_{4,5}^\top \mathbf{y}) \\
&= \frac{4}{3} \begin{bmatrix} \sqrt{\frac{1}{2}} & \sqrt{\frac{1}{2}} \end{bmatrix} \begin{bmatrix} 1 & -\frac{1}{2} \\ -\frac{1}{2} & 1 \end{bmatrix} \begin{bmatrix} \sqrt{\frac{1}{2}} \\ \sqrt{\frac{1}{2}} \end{bmatrix} = \frac{2}{3} < 1
\end{aligned}
$$

Thus, ADAPTIVE-SAMPLING will enter an infinite while loop and never terminate.

We note that greedy achieves the optimal solution by first selecting a feature from $\mathbf{x}_4, \mathbf{x}_5, \mathbf{x}_6$ in the first iteration and then selecting the second feature from $\mathbf{x}_1, \mathbf{x}_2, \mathbf{x}_3$.

# B  Notions of Approximate Submodularity

In this section, we discuss related work on notions of non-submodularity and their theoretical guarantees on choosing a set of size $k$ to comprise the solution set. Our definition differs from these notions in three aspects and allows for parallelization. Specifically, 1) we bound the marginal contribution of the objective function $f$ and not just the function value and 2) we consider the marginal contribution of sets of elements instead of a singleton and 3) we allow the flexibility of being bound by two different submodular functions. These alterations are necessary for the proof of our low-adaptivity algorithm.

Krause et al. [KC10] define *approximate submodularity* with parameter $\epsilon \geq 0$ as functions that satisfy an additive approximate diminishing returns property, i.e. $\forall S \subseteq T \subseteq N \setminus a$ it holds that $f_S(a) \geq f_T(a) - \epsilon$. SDS$_{\text{MA}}$ applied to functions with this additive property inherits an additive guarantee of $f(S) \geq (1 - 1/e)f(O) - k\epsilon$.

Das and Kempe [DK11] define the *submodularity ratio* with parameter $\gamma \geq 0$ to quantify how close a function is to submodularity, where $\gamma = \min_{S,A} \frac{\sum_{a \in A} f_S(a)}{f_S(A)}$. Elenberg et al. [EKD$^+$18] extend their work and lower bound the submodularity ratio using strong concavity and smoothness parameters for generalized linear models. SDS$_{\text{MA}}$ applied to functions with this property inherits a guarantee of $f(S) \geq (1 - 1/e^\gamma)f(O)$. Because $\gamma$ is difficult to compute on a real dataset (only possible using brute force), Bian et al. [BBKT17] introduce the Greedy submodularity ratio $\gamma^G = \min_{A:|A|=k,S^t} \frac{\sum_{a \in A} f_{S^t}(a)}{f_{S^t}(A)}$, where $S^t$ is the set chosen by the greedy algorithm at step $t$.

For multiplicative bounds, Horel et al. [HS16] define $\epsilon$-*approximately submodular* functions where $f$ is approximately submodular if there exists a submodular function $g$ s.t. $(1 - \epsilon)g(S) \leq f(S) \leq (1 + \epsilon)g(S), \forall S \subseteq N$. In this definition, the function is approximated pointwise by a submodular function, not its marginals as in differential submodularity. Gupta et al. [GPB18] define a similar property on the marginals of the function where $f$ is $\delta$-*approximately submodular* if there exists a submodular function $g$ s.t. $(1 - \delta)g_S(a) \leq f_S(a) \leq (1 + \delta)g_S(a), \forall S \subseteq N, a \notin S$. Differential submodularity generalizes this definition so that the functions that bound the objective can differ. This is necessary in cases where the objective function contains a diversity factor.

# C  Relationship to PRAM

The PRAM model is a generalization of the RAM model with parallelization. It represents an idealized model that can execute instructions in parallel with any number of processors in a shared memory machine. In this framework, the notion of depth is closely related to the one of adaptivity that we discuss in this paper. The *depth* of a PRAM model is the number of parallel steps in an algorithm or the longest chain of dependencies. The area of designing low-depth algorithms have been extensively studied. Our results extend to the PRAM model, similarly to the results of the original adaptive sampling algorithm for submodular maximization. For more detail, please see Appendix A.2.2 of [BS18b].

# D  Bayesian Experimental Design Details

In Bayesian experimental design, we would like to select a set of experiments to optimize some statistical criterion. Specifically, the Bayesian A-optimality criterion is used to maximally reduce the variance in the posterior distribution over the parameters.

More formally, let $n$ experimental stimuli comprise the matrix $\mathbf{X} \in \mathbb{R}^{d \times n}$, where each experimental stimuli $\mathbf{x}_i \in \mathbb{R}^d$ is a column in $\mathbf{X}$. We can select a set $S \subseteq \mathcal{N}$ of stimuli and denote this as $\mathbf{X}_S \in \mathbb{R}^{d \times |S|}$. Let $\theta \in \mathbb{R}^d$ be the parameter vector in the linear model $\mathbf{y}_S = \mathbf{X}_S^T \theta + \mathbf{w}$, where $\mathbf{w} \sim \mathcal{N}(0, \sigma^2 \mathbf{I})$ is noise from a Gaussian distribution, $\mathbf{y}_S$ is the vector of dependent variables, and $\theta \sim \mathcal{N}(0, \mathbf{\Lambda}^{-1}), \mathbf{\Lambda} = \beta^2 \mathbf{I}$ is the prior that takes the form of an isotropic Gaussian. Then,

$$\begin{bmatrix} \mathbf{y}_S \\ \theta \end{bmatrix} \sim \mathcal{N}(0, \mathbf{\Sigma}), \mathbf{\Sigma} = \begin{bmatrix} \sigma^2 \mathbf{I} + \mathbf{X}_S^T \mathbf{\Lambda}^{-1} \mathbf{X}_S & \mathbf{X}_S^T \mathbf{\Lambda}^{-1} \\ \mathbf{\Lambda}^{-1} \mathbf{X}_S & \mathbf{\Lambda}^{-1} \end{bmatrix}$$

which implies $\mathbf{\Sigma}_{\theta|\mathbf{y}_S} = (\mathbf{\Lambda} + \sigma^{-2} \mathbf{X}_S \mathbf{X}_S^T)^{-1}$.

Now, we can define our A-optimality objective as

$$f_{\texttt{A-opt}}(S) = \mathrm{Tr}(\Sigma_\theta) - \mathrm{Tr}(\boldsymbol{\Sigma}_{\theta|\mathbf{y}_s} = \mathrm{Tr}(\boldsymbol{\Lambda}^{-1}) - \mathrm{Tr}((\boldsymbol{\Lambda} + \sigma^{-2}\mathbf{X}_S\mathbf{X}_S^T)^{-1})) \tag{4}$$

To regularize for diverse experiments, we can formulate the problem as follows

$$\max_{S:|S|\leq k} f_{\texttt{A-div}}(S) = f_{\texttt{A-opt}}(S) + d(S),$$

where $d : 2^N \to \mathbb{R}_+$ is a "diverse" submodular function promoting regularization.

Krause et al. [KSG08] has shown that the Bayesian A-optimality objective is not submodular and Bian et al. [BBKT17] has shown that submodularity ratio of the objective can be lower bounded. With the traditional greedy algorithm, we get a $1 - 1/e^\gamma$ approximation guarantee, where $\gamma \geq \frac{\beta^2}{\|\mathbf{X}\|^2(\beta^2+\sigma^{-2}\|\mathbf{X}\|^2)}$ [BBKT17].

# E    Missing Proofs from Section 3

## E.1    Proof of Corollary 7

*Proof.* In the case where there is no diversity regularization term, the concavity and smoothness parameters correspond to the sparse eigenvalues of the covariance matrix, i.e., $m_k = \lambda_{min}(k)$ and $M_k = \lambda_{max}(k)$ [EKD+18].

Thus, by Theorem 6, we can also write the bounds for $f_S(A)$ in terms of eigenvalues $\frac{\lambda_{min}(s)}{\lambda_{max}(t)}\tilde{f}_S(A) \leq f_S(A) \leq \frac{\lambda_{max}(s)}{\lambda_{min}(t)}\tilde{f}_S(A)$, where $\tilde{f}_S(A) = \sum_{a\in A} f_S(a)$. With $g_S(A) = \frac{\lambda_{min}(s)}{\lambda_{max}(t)}\tilde{f}_S(A)$ and $h_S(A) = \frac{\lambda_{max}(s)}{\lambda_{min}(t)}\tilde{f}_S(A)$, we get that that the objective is a $(\frac{\lambda_{min}(t)}{\lambda_{max}(t)})^2$-differentially submodular function. Since $2k \geq t$, we get the desired result.

In the case where there is a diversity regularization term in the objective $f_{\texttt{div}}(S) = \ell_{\texttt{reg}}(\mathbf{w}^{(S)})+d(S)$, we have

$$\frac{\lambda_{min}(s)}{\lambda_{max}(t)}\tilde{f}_S(A) + d_S(A) \leq (f_{\texttt{div}})_S(A) \leq \frac{\lambda_{max}(s)}{\lambda_{min}(t)}\tilde{f}_S(A) + d_S(A).$$

With $g_S(A) = \frac{\lambda_{min}(s)}{\lambda_{max}(t)}\tilde{f}_S(A) + d_S(A)$ and $h_S(A) = \frac{\lambda_{max}(s)}{\lambda_{min}(t)}\tilde{f}_S(A) + d_S(A)$, we get that $g_S(A)/h_S(A) \geq \frac{\lambda_{min}(s)\lambda_{min}(t)}{\lambda_{max}(s)\lambda_{max}(t)} \geq (\frac{\lambda_{min}(t)}{\lambda_{max}(t)})^2$ since $d_S(A) \geq 0$. Since $2k \geq t$, this concludes the proof. □

**Remark 13.** *Since $\lambda_{max}(s) = 1$, the upper bound of $\frac{\lambda_{max}(s)}{\lambda_{min}(t)}\tilde{f}_S(A) \leq f_S(A)$ is consistent with the result in Lemma 2.4 from Das and Kempe [DK11] that shows that the weak submodularity ratio can be lower bounded by $\lambda_{min}$.*

## E.2    Proof of Corollary 8

*Proof.* The first portion of the proof relies on the result from Elenberg et al. [EKD+18]. In general, log-likelihood functions of generalized linear models (GLMs) are not RSC/RSM, but their result shows that log-likelihood objectives are RSC/RSM with parameters $m$ and $M$ under mild conditions of the feature matrix.

Our result follows directly from Theorem 6. The case where there is a diversity regularization term then follows similarly as for Corollary 7. □

## E.3    Proof of Corollary 9

*Proof.* In the case where there is no diversity regularization term, we can upper bound the submodularity ratio to prove differential submodularity.

We first lower bound the marginal contribution of a set $A$ to $S$, $(f_{\texttt{A-opt}})_S(A)$ and then upper bound the marginal contribution of one element $a$ to the set $S$, $(f_{\texttt{A-opt}})_S(a)$.

$$
\begin{aligned}
f_S(A) &= \sum_{i=1}^{d} \frac{1}{\beta^2 + \sigma^{-2}\sigma_i^2(\mathbf{X}_S)} - \sum_{j=1}^{d} \frac{1}{\beta^2 + \sigma^{-2}\sigma_i^2(\mathbf{X}_{S\cup A})} \\
&= \sum_{i=1}^{d} \frac{\sigma^{-2}[\sigma_i^2(\mathbf{X}_{S\cup A}) - \sigma_i^2(\mathbf{X}_S)]}{(\beta^2 + \sigma^{-2}\sigma_i^2(\mathbf{X}_S))(\beta^2 + \sigma^{-2}\sigma_i^2(\mathbf{X}_{S\cup A}))} \\
&\geq (\beta^2 + \sigma^{-2}\sigma_{max}^2(\mathbf{X}))^{-2} \sum_{i=1}^{d} \sigma^{-2}[\sigma_i^2(\mathbf{X}_{S\cup A}) - \sigma_i^2(\mathbf{X}_S)] \\
&= (\beta^2 + \sigma^{-2}\|\mathbf{X}\|^2)^{-2} \sum_{i=1}^{d} \sigma^{-2}[\lambda_i(\mathbf{X}_{S\cup A}\mathbf{X}_{S\cup A}^T) - \lambda_i(\mathbf{X}_S\mathbf{X}_S^T)] \\
&= (\beta^2 + \sigma^{-2}\|\mathbf{X}\|^2)^{-2}\sigma^{-2}[\mathrm{Tr}(\mathbf{X}_{S\cup A}\mathbf{X}_{S\cup A}^T) - \mathrm{Tr}(\mathbf{X}_S\mathbf{X}_S^T)] \\
&= (\beta^2 + \sigma^{-2}\|\mathbf{X}\|^2)^{-2}\sigma^{-2}[\mathrm{Tr}(\mathbf{X}_S\mathbf{X}_S^T + \mathbf{X}_A\mathbf{X}_A^T) - \mathrm{Tr}(\mathbf{X}_S\mathbf{X}_S^T)] \\
&= (\beta^2 + \sigma^{-2}\|\mathbf{X}\|^2)^{-2}\sigma^{-2}\,\mathrm{Tr}(\mathbf{X}_A\mathbf{X}_A^T) \\
&= (\beta^2 + \sigma^{-2}\|\mathbf{X}\|^2)^{-2} \sum_{a\in A} \sigma^{-2}\,\mathrm{Tr}(\mathbf{x}_a\mathbf{x}_a^T) \\
&= (\beta^2 + \sigma^{-2}\|\mathbf{X}\|^2)^{-2} \sum_{a\in A} \|\mathbf{x}_a\|^2 \\
&= \sigma^{-2}(\beta^2 + \sigma^{-2}\|\mathbf{X}\|^2)^{-2}|A| \qquad (5)
\end{aligned}
$$

$$
\begin{aligned}
\sum_{a\in A} f_S(a) &= \sum_{a\in A}\sum_{i=1}^{d} \frac{1}{\beta^2 + \sigma^{-2}\sigma_i^2(\mathbf{X}_S)} - \sum_{j=1}^{d} \frac{1}{\beta^2 + \sigma^{-2}\sigma_i^2(\mathbf{X}_{S\cup a})} \\
&\leq \sum_{a\in A} \frac{1}{\beta^2 + \sigma^{-2}\sigma_d^2(\mathbf{X}_S)} - \frac{1}{\beta^2 + \sigma^{-2}\sigma_1^2(\mathbf{X}_{S\cup a})} \\
&\leq \sum_{a\in A} \frac{1}{\beta^2} - \frac{1}{\beta^2 + \sigma^{-2}\sigma_1^2(\mathbf{X}_{S\cup a})} \\
&= \sum_{a\in A} \frac{\sigma^{-2}\sigma_1^2(\mathbf{X}_{S\cup a})}{\beta^2(\beta^2 + \sigma^{-2}\sigma_1^2(\mathbf{X}_{S\cup a}))} \\
&= |A|\frac{\sigma^{-2}\|\mathbf{X}\|^2}{\beta^2(\beta^2 + \sigma^{-2}\|\mathbf{X}\|^2)} \qquad (6)
\end{aligned}
$$

Combining (5) and (6), yields

$$
\frac{\sum_{a\in A} f_S(a)}{f_S(A)} \leq \frac{|A|\frac{\sigma^{-2}\|\mathbf{X}\|^2}{\beta^2(\beta^2 + \sigma^{-2}\|\mathbf{X}\|^2)}}{\sigma^{-2}(\beta^2 + \sigma^{-2}\|\mathbf{X}\|^2)^{-2}|A|} = \frac{\|\mathbf{X}\|^2(\beta^2 + \sigma^{-2}\|\mathbf{X}\|^2)}{\beta^2}.
$$

Bian et al. [BBKT17] showed that the submodularity ratio can be lower bounded by $\frac{\beta^2}{\|\mathbf{X}\|^2(\beta^2 + \sigma^{-2}\|\mathbf{X}\|^2)}$.

With $g_S(A) = \frac{\beta^2}{\|\mathbf{X}\|^2(\beta^2 + \sigma^{-2}\|\mathbf{X}\|^2)}\tilde{f}_S(A)$ and $h_S(A) = \frac{\|\mathbf{X}\|^2(\beta^2 + \sigma^{-2}\|\mathbf{X}\|^2)}{\beta^2}\tilde{f}_S(A)$, we get that that the objective is a $\gamma^2$-differentially submodular function where $\gamma = \frac{\beta^2}{\|\mathbf{X}\|^2(\beta^2 + \sigma^{-2}\|\mathbf{X}\|^2)}$.

In the case where there is a diversity regularization term in the objective, we can follow similar reasoning from Corollary 7 to conclude the proof. $\square$

# F Extension to $R^2$ Objective

## F.1 Goodness of Fit

We introduce the formal definition of the $R^2$ objective function, which is widely used to measure goodness of fit in statistical applications.

**Definition 14.** *[JW04] Let $S \subseteq N$ be a set of variables $\mathbf{X}_S$ and a linear predictor $\hat{\mathbf{y}} = \sum_{i \in S} \beta_i \mathbf{X}_i$ of $\mathbf{y}$, the squared multiple correlation is defined as*

$$R^2(S) = \frac{Var(\mathbf{y}) - \mathbb{E}[(\mathbf{y} - \hat{\mathbf{y}})^2]}{Var(\mathbf{y})}$$

*where $\beta_i = (\mathbf{C}_S)^{-1} \mathbf{b}_S$ for $i \in S$.*

We assume that the predictor random variables are normalized to have mean 0 and variance 1, so we can simplify the definition above to $R^2(S) = 1 - \mathbb{E}[(\mathbf{y} - \hat{\mathbf{y}})^2]$. Thus, we can rephrase the definition as $R^2(S) = \mathbf{b}_S^T (\mathbf{C}_S)^{-1} \mathbf{b}_S$. [JW04].

## F.2 Feature Selection

**Objective.** For a response variable $\mathbf{y} \in \mathbb{R}^d$, the objective is the maximization of the $R^2$ goodness of fit for $\mathbf{y}$ given the feature set $S$:

$$f(S) = R^2(S) = \mathbf{b}_S^T (\mathbf{C}_S)^{-1} \mathbf{b}_S$$

where $\mathbf{b}$ corresponds to the covariance between $\mathbf{y}$ and the predictors.

To define the marginal contribution of a set $A$ to the set $S$ of the $R^2$ objective function, we can write $R_S^2(A) = (\mathbf{b}_A^S)^T (\mathbf{C}_A^S)^{-1} \mathbf{b}_A^S$, where $\mathbf{b}^S$ is the covariance vector corresponding to the residuals of $i \in A$ to $S$, i.e. $\{\text{Res}(\mathbf{x}_1, \mathbf{X}_S), \text{Res}(\mathbf{x}_2, \mathbf{X}_S), \dots, \text{Res}(\mathbf{x}_n, \mathbf{X}_S)\}$ and $\mathbf{C}_A^S$ is the covariance matrix corresponding to the residuals. The marginal contribution of an element is $R_S^2(a) = (\mathbf{b}_a^S)^T \mathbf{b}_a^S$.

**Lemma 15.** *The feature selection objective defined by $f(S) = R^2(S)$ is a $\frac{\lambda_{min}(\mathbf{C}_A^S)}{\lambda_{max}(\mathbf{C}_A^S)}$-differentially submodular function such that for all $S, A \subseteq N$,*

$$g_S(A) = \frac{1}{\lambda_{max}(\mathbf{C}_A^S)} \tilde{f}_S(A) \leq f_S(A) \leq \frac{1}{\lambda_{min}(\mathbf{C}_A^S)} \tilde{f}_S(A) = h_S(A),$$

*where $\tilde{f}_S(A) = \sum_{a \in A} f_S(a)$.*

*Proof.* The marginal contribution of set $A$ to set $S$ of the feature selection objective function is defined as $R_S^2(A) = (\mathbf{b}_A^S)^T (\mathbf{C}_A^S)^{-1} \mathbf{b}_A^S$. Because we know that $(\mathbf{C}_A^S)^{-1}$ is a symmetric matrix, we can upper and lower bound the marginals using the eigenvalues of $(\mathbf{C}_A^S)^{-1}$.

$$
\begin{aligned}
\frac{1}{\lambda_{max}(\mathbf{C}_A^S)} \sum_{a \in A} f_S(a) &= \frac{1}{\lambda_{max}(\mathbf{C}_A^S)} (\mathbf{b}_A^S)^T \mathbf{b}_A^S \\
&= \lambda_{min}((\mathbf{C}_A^S)^{-1})(\mathbf{b}_A^S)^T \mathbf{b}_A^S \\
&\leq (\mathbf{b}_A^S)^T (\mathbf{C}_A^S)^{-1} \mathbf{b}_A^S \\
&= f_S(A) \\
&\leq \lambda_{max}((\mathbf{C}_A^S)^{-1})(\mathbf{b}_A^S)^T \mathbf{b}_A^S \\
&\leq \frac{1}{\lambda_{min}(\mathbf{C}_A^S)} (\mathbf{b}_A^S)^T \mathbf{b}_A^S \\
&= \frac{1}{\lambda_{min}(\mathbf{C}_A^S)} \sum_{a \in A} f_S(a)
\end{aligned}
$$

By letting $\tilde{f}_S(A) = \sum_{a \in A} f_S(a)$, we complete the proof and show that the marginals can be bounded by modular functions. $\square$

**Remark 16.** *This is a more general form of Lemma 3.3 from Das and Kempe [DK11]. Our result is on the marginals of $f$ and reduces to their result for $S = \emptyset$.*

**Remark 17.** *If $\lambda_{min} = \lambda_{max}$, the matrix has one eigenvalue of multiplicity greater than 1 and the covariance matrix is a multiple of the identity matrix. This implies the set of predictors is uncorrelated and that the objective function for feature selection is submodular. Otherwise, we have $\alpha = \frac{\lambda_{min}}{\lambda_{max}} < 1$.*

## G   Additional Algorithm Detail

We briefly discuss how to estimate the expectations that appear in the algorithm. We also discuss how to estimate OPT and differential submodularity parameter $\alpha$. For the full algorithm and details, see Appendix A.C.2 in [BS18b].

Since we do not know the value of $\mathbb{E}_{R \sim \mathcal{U}(X)}[f_S(R)]$, we can estimate it with $m$ samples. We first randomly select sets uniformly $R_1, R_2, ...R_m \sim \mathcal{U}(X)$ and compute $f_S(R_i)$. Then we can average these calculations to estimate the expected marginal contribution. Balkanski et al. discuss the number of samples needed to bound the error of these estimates [BS18b]. Specifically, with $m = \frac{1}{2}(\frac{\text{OPT}}{\epsilon})^2 \log(\frac{2}{\delta})$, then with probability at least $1 - \delta$,

$$\left| \left( \frac{1}{m} \sum_{i=1}^m f(S \cup R_i) - f(S) \right) - \mathbb{E}_{R \sim \mathcal{U}(X)}[f_S(R)] \right| \leq \epsilon$$

.

Similarly, let $m = \frac{1}{2}(\frac{\text{OPT}}{\epsilon})^2 \log(\frac{2}{\delta})$, then for all $S \subseteq N$ and $a \in N$, with probability at least $1 - \delta$ over samples $R_1, ..., R_m$,

$$\left| \left( \frac{1}{m} \sum_{i=1}^m f(S \cup R_i \cup \{a\}) - f(S \cup R_i \backslash \{a\}) \right) - \mathbb{E}_{R \sim \mathcal{U}(X)}[f_{S \cup R \backslash \{a\}}(a)] \right| \leq \epsilon.$$

Thus, for $m = n(\frac{\text{OPT}}{\epsilon})^2 \log(\frac{2n}{\delta})$ total samples in one round, we can get $\epsilon$-estimates for marginal contributions. For proof details, see Lemma 6 in [BS18b]. We note that in practice, we observe comparable terminal values compared to the greedy algorithm even with much fewer number of samples.

To estimate OPT, we can "guess" the value of OPT and run several of these guesses in parallel. One can set $\text{OPT} \in \{(1 + \epsilon)^i \max_{a \in N} f(a) : i \in \left[ \frac{\ln(n)}{\epsilon} \right] \}$. One such value $i$ is guaranteed to be a $(1 - \epsilon)$-approximation to OPT [BS18b]. Similarly for the differential submodularity parameter $\alpha$, we can guess values so that $\alpha \in \{(1 + \epsilon)^i : i \in \left[ \frac{\ln(n)}{\epsilon} \right] \}$ and run these guesses in parallel. In practice, we found that the algorithm performance was not very sensitive to parameter estimates and we could observe comparable terminal value without much parameter tuning.

## H   Proof of Theorem 10 for DASH

We first prove several lemmas before proving the theorem.

### H.1   Proofs of Lemmas Leading to Theorem 10

We first begin by proving the following lemma to bound the marginal contribution of the optimal set to the solution set.

**Lemma 18.** *Let $R_i \sim \mathcal{U}(X)$ be the random set at iteration $i$ of $\text{DASH}(N, S, r, \delta)$. For all $S \subseteq N$ and $r, \rho > 0$, if the algorithm has not terminated after $\rho$ iterations, then*

$$\mathbb{E}_{R_i}[f_{S \cup (\cup_{i=1}^\rho R_i)}(O)] \geq (1 - \frac{\rho}{r})(f(O) - f(S)) \tag{7}$$

Using Lemma 18, we can complete the proof for Lemma 19.

*Proof.*

$$\mathbb{E}_{R_i}[f_{S \cup (\cup_{i=1}^{\rho} R_i)}(O)] = \mathbb{E}_{R_i}[f_S(O \cup (\cup_{i=1}^{\rho} R_i))] - \mathbb{E}_{R_i}[f_S(\cup_{i=1}^{\rho} R_i)]$$

$$\geq f(O) - f(S) - \frac{1}{\alpha} \sum_{i=1}^{\rho} \mathbb{E}_{R_i}[f_S(R_i)]$$

$$\geq f(O) - f(S) - \frac{1}{\alpha} \alpha \rho (\frac{1-\epsilon}{r}(f(O) - f(S))$$

$$\geq (1 - \frac{\rho}{r})(f(O) - f(S))$$

where the first inequality follows from monotonicity and differential submodularity and the second inequality follows from the `while` loop in DASH. $\square$

**Lemma 19.** *For each iteration of* DASH *and for all* $S \subseteq N$ *and* $\epsilon > 0$, *if* $r \geq 20 \rho \epsilon^{-1}$ *then the marginal contribution of the elements of* $X_\rho$ *that survive* $\rho$ *iterations satisfy*

$$f_S(X_\rho) \geq \frac{\alpha^2}{r}(1 - \epsilon)(f(O) - f(S))$$

*Proof.* We want to show a bound on the marginal contribution of the elements that survive $\rho$ iterations of the algorithm. To prevent the propagation of the $\alpha$ factor, we upper and lower bound $f$ by two submodular functions $h$ and $g$ for our analysis, excluding the queries made by the algorithm.

Let $O = \{o_1, \ldots, o_k\}$ be the optimal solutions of $f$ and $O_l = \{o_1, \ldots, o_l\}$ be a subset of the optimal elements in some arbitrary order. However, we define the thresholds in terms of submodular function $h$. Then we define

$$\Delta_l := \mathbb{E}_{R_i}[h_{S \cup O_{l-1} \cup (\cup_{i=1}^{\rho} R_i) \setminus \{o_l\}}(o_l)] \tag{8}$$

$$\Delta := \frac{1}{k} \mathbb{E}_{R_i}[h_{S \cup (\cup_{i=1}^{\rho} R_i)}(O)] \tag{9}$$

Let $r \geq \frac{20\rho}{\epsilon}$. Let $T$ be the set of elements surviving $\rho$ iterations in $O$, $T \subseteq X_\rho$, $T \subseteq O$, where

$$T = \{o_l | \Delta_l \geq (1 - \frac{\epsilon}{4})\Delta\} \tag{10}$$

For $o_l \in T$ and using differential submodularity properties,

$$\mathbb{E}_{R_i}[f_{S \cup (\cup_{i=1}^{\rho} R_i \setminus \{o_l\})}(o_l)] \geq \mathbb{E}_{R_i}[g_{S \cup (\cup_{i=1}^{\rho} R_i \setminus \{o_l\})}(o_l)]$$

$$\geq \mathbb{E}_{R_i}[\alpha h_{S \cup (\cup_{i=1}^{\rho} R_i \setminus \{o_l\})}(o_l)]$$

$$\geq \alpha \mathbb{E}_{R_i}[h_{S \cup O_{l-1} \cup (\cup_{i=1}^{\rho} R_i \setminus \{o_l\})}(o_l)]$$

$$\geq \alpha (1 - \frac{\epsilon}{4})\Delta \qquad \text{Definition (10)}$$

$$\geq \frac{\alpha}{k}(1 - \frac{\epsilon}{4})\mathbb{E}_{R_i}[f_{S \cup (\cup_{i=1}^{\rho} R_i)}(O)]$$

$$\geq \frac{\alpha}{k}(1 - \frac{\epsilon}{4})(1 - \frac{\rho}{r})(f(O) - f(S)) \qquad \text{Lemma 18}$$

$$\geq \frac{\alpha}{k}(1 + \frac{\epsilon}{2})(1 - \epsilon)(f(O) - f(S)) \qquad r \geq \frac{20\rho}{\epsilon} \tag{11}$$

which shows that elements in $T$ survive the elimination process (as they are not filtered out from set $X$ in the algorithm definition).

Now we complete the proof by showing $f_S(T)$ is bounded by $\frac{\alpha^2}{r}(1 - \epsilon)(f(O) - f(S))$ which effectively terminates the algorithm.

Similar to the result in Lemma 2 of Balkanski et al. [BRS19a], from properties of submodularity of $g$ and $h$, we have

$$\sum_{o_l \in T} \Delta_l \geq k \frac{\epsilon}{4} \Delta \tag{12}$$

By submodularity,

$$
\begin{aligned}
f_S(T) &\geq g_S(T) \\
&\geq \alpha h_S(T) \\
&\geq \alpha \sum_{o_l \in T} h_{S \cup O_{l-1}}(o_l) \\
&\geq \alpha \sum_{o_l \in T} \mathbb{E}[h_{S \cup O_{l-1} \cup (\cup_{i=1}^\rho R_i \setminus \{o_l\})}(o_l)] \\
&= \alpha \sum_{o_l \in T} \Delta_l && \text{Definition (8)} \\
&\geq (1-\delta)k\Delta\frac{\epsilon}{4} && \text{from (12)} \\
&\geq \alpha \frac{\epsilon}{4} \mathbb{E}_{R_i}[f_{S \cup (\cup_{i=1}^\rho R_i)}(O)]
\end{aligned}
$$

where the second and third inequalities follow from properties of submodularity. Finally,

$$
\begin{aligned}
f_S(X_\rho) &\geq f_S(T) && \text{monotonicity} \\
&= \alpha(\frac{\epsilon}{4})(1 - \frac{\rho}{r})(f(O) - f(S)) && \text{Definition (9)} \\
&\geq \frac{\alpha^2}{r}(1-\epsilon)(f(O) - f(S)) && r \geq \frac{20\rho}{\epsilon}
\end{aligned}
$$

$\square$

We now present a lemma for the termination of the algorithm in $\mathcal{O}(\log n)$ rounds.

**Lemma 20.** *Let $X_i$ and $X_{i+1}$ be the sets of surviving elements at the start and end of iteration $i$ of the* `while` *loop of* DASH. *For all $S \subseteq N$ and $r, i, \epsilon > 0$, if the algorithm does not terminate at iteration $i$, then*

$$
|X_{i+1}| < \frac{|X_i|}{1 + \epsilon/2}
$$

*Proof.* We consider $R_i \cap X_{i+1}$ to bound the number of surviving elements in $X_{i+1}$. To prevent the propagation of the $\alpha$ factor, we can bound the function $f$ by its submodular bounds.

$$
\begin{aligned}
\mathbb{E}[f_S(R_i \cap X_{i+1})] &\geq \mathbb{E}[g_S(R_i \cap X_{i+1})] \\
&\geq \alpha \mathbb{E}[\sum_{a \in R_i \cap X_{i+1}} h_{S \cup (R_i \cap X_{i+1} \setminus a)}(a)] \\
&\geq \alpha \mathbb{E}[\sum_{a \in X_{i+1}} \mathbb{1}_{a \in R_i} \cdot h_{S \cup (R_i \setminus a)}(a)] \\
&= \alpha \sum_{a \in X_{i+1}} \mathbb{E}[\mathbb{1}_{a \in R_i} \cdot h_{S \cup (R_i \setminus a)}(a)] \\
&= \alpha \sum_{a \in X_{i+1}} \mathbb{P}[a \in R_i] \cdot \mathbb{E}[h_{S \cup (R_i \setminus a)}(a) | a \in R_i] \\
&\geq \alpha \sum_{a \in X_{i+1}} \mathbb{P}[a \in R_i] \cdot \mathbb{E}[h_{S \cup (R_i \setminus a)}(a)] \\
&\geq \alpha \sum_{a \in X_{i+1}} \mathbb{P}[a \in R_i] \cdot \mathbb{E}[f_{S \cup (R_i \setminus a)}(a)] \\
&\geq \alpha \sum_{a \in X_{i+1}} \mathbb{P}[a \in R_i] \cdot \frac{\alpha}{k}(1 + \epsilon/2)(1-\epsilon)(f(O) - f(S)) \\
&= \alpha^2 \frac{|X_{i+1}|}{|X_i|} \frac{k}{r} \cdot \frac{1}{k}(1 + \epsilon/2)(1-\epsilon)(f(O) - f(S)) \\
&= \alpha^2 \frac{|X_{i+1}|}{r|X_i|}(1 + \epsilon/2)(1-\epsilon)(f(O) - f(S)) && \text{(13)}
\end{aligned}
$$

where the first and fourth inequalities are due to differential submodularity.

Since the elements are discarded from the `while` loop of the algorithm, we can bound $\mathbb{E}[f_S(R_i \cap X_{i+1})]$ using monotonicity so that

$$\mathbb{E}[f_S(R_i \cap X_{i+1})] \leq \mathbb{E}[f_S(R_i)] < \alpha^2(1-\epsilon)(f(O) - f(S))/r. \tag{14}$$

Combining (13) and (14) yields

$$(1-\epsilon)(f(O) - f(S))/r > \frac{1}{\alpha^2}\mathbb{E}[f_S(R_i)] \geq \frac{|X_{i+1}|}{r|X_i|}(1+\epsilon/2)(1-\epsilon)(f(O) - f(S))$$

We can conclude that $|X_{i+1}| < |X_i|/(1+\epsilon/2)$ by simplifying that above inequality. $\qquad\square$

**Lemma 21.** *For all $S \subseteq N$, if $r \geq 20\epsilon^{-1}\log_{(1+\epsilon/2)}(n)$ then $\mathrm{DASH}(N, S, r, \delta)$ terminates after at most $\mathcal{O}(\log n)$ rounds.*

*Proof.* If the algorithm has not terminated after $\log_{1+\epsilon/2}(n)$ rounds, then, by Lemma 21, at most $k/r$ elements survived $\rho = \log_{1+\epsilon/2}(n)$ iterations. By Lemma 19, the set of surviving elements satisfies $f_S(X_\rho) \geq \frac{\alpha^2}{r}(1-\epsilon)(f(O) - f(S))$. Since there are only $k/r$ surviving elements, $R = X_\rho$ and

$$f_S(R) = f_S(X_\rho) \geq \frac{\alpha^2}{r}(1-\epsilon)(f(O) - f(S))$$

$\qquad\square$

## H.2  Proof of Theorem 10

*Proof.* We prove the theorem by induction. From Lemma 21, we know

$$f(S_i) \geq f(S_{i-1}) + \alpha^2\frac{1-\epsilon}{r}(f(O) - f(S_{i-1}))$$

By subtracting $f(O)$, this is equivalent to

$$f(S_i) - f(O) \geq (1 - \alpha^2\frac{1-\epsilon}{r})[f(S_{i-1}) - f(O)]$$

By induction and rearranging, we have

$$\begin{aligned} f(S_i) - f(O) &\geq (1 - \alpha^2\frac{1-\epsilon}{r})^i(-f(O)) \\ &= -(1 - \alpha^2\frac{1-\epsilon}{r})^i f(O) \end{aligned}$$

By setting $i = r$ and rearranging, we have

$$\begin{aligned} f(S) &\geq (1 - (1 - \alpha^2\frac{1-\epsilon}{r})^r)f(O) \\ &\geq (1 - e^{-\alpha^2(1-\epsilon)})f(O) \\ &\geq (1 - 1/e^{\alpha^2} - \alpha^2\epsilon)f(O) \end{aligned}$$

$\qquad\square$

# I  Additional Detail for Experiments

## I.1  Experimental Setup

All algorithms were implemented in Python 3.6. Experiments on third-party datasets were conducted on AWS EC2 C4 with 2.9 GHz Intel Xeon E5-2666 v3 Processors on 16 or 36 cores. Experiments on synthetic datasets ran on 3.1 GHz Intel Core i7 processors on 8 cores.

## I.2 Datasets

This section details the generation of synthetic data and the real clinical and biological datasets we used for experiments. D1 and D2 are used in linear regression and Bayesian experimental design applications and D3 and D4 are used in logistic regression for classification applications.

- **D1: Synthetic Dataset for Regression and Experimental Design.** We generated 500 features by sampling from a multivariate normal distribution. Each feature is normalized to have mean 0 and variance 1. Furthermore, features have a covariance of 0.4 to guarantee differential submodularity. To generate our response variable $\mathbf{y}$, we sample the coefficient $\beta \sim \mathcal{U}(-2, 2)$ for a subset of size 100 from the feature set and compute $\mathbf{y}$ after adding a small noise term to the coefficients. Our goal is to select features that have coefficients of large magnitude and accurately predict the response variable $\mathbf{y}$.

  We generated the dataset for experimental design similarly. We generated 256 features and 1024 samples by sampling from a multivariate normal distribution. Each feature is normalized to have mean 0 and variance 1. Features have a covariance of 0.8. Each row is then normalized to have $\ell_2$ norm of 1;

- **D2: Clinical Dataset for Regression and Experimental Design.** We used a publicly available dataset with 53,500 samples from 74 patients with 385 features and want to select a smaller set of features that can accurately predict the location on the axial axis from an image of the brain slice. For experimental design, we sample 1000 rows from the dataset to comprise our sample space and normalize rows to have $\ell_2$ norm of 1;

- **D3: Synthetic Dataset for Classification.** We generated a synthetic dataset for logistic regression using a similar methodology as the synthetic regression dataset. We select a set of 50 true support features from a set of 200 and generate the coefficients using $\mathcal{U}(-2, 2)$. However, instead of a numerical response variable, we create a two-class classification problem by transforming the continuous $\mathbf{y}$ into probabilities and assigning the class label using a threshold of 0.5. The goal is to select features to perform binary classification on the synthetic dataset by using the log likelihood objective;

- **D4: Biological Dataset for Classification.** We used clinical data that contains the presence or absence of 2,500 genes in 10,633 samples from various patients. In this 5-class multi-classification problem, we want to select a small set of genes that can accurately predict the site of cancer metastasis (spleen, colon, parietal peritoneum, mesenteric lymph node, and intestine).

## I.3 Benchmarks

We compared DASH to these algorithms:

- **RANDOM.** In one round, this algorithm randomly selects $k$ features to create the solution set;

- **TOP-$k$.** In one round, this algorithm selects the $k$ features whose individual objective value is largest;

- **SDS$_{\text{MA}}$.** This uses the traditional greedy algorithm to select elements with the largest marginal contribution at each round [KC10]. In each round, the algorithm adds one element to the solution set;

- **Parallel SDS$_{\text{MA}}$.** To compare parallel runtime between DASH and greedy, we also implemented a parallelized version of the SDS$_{\text{MA}}$ algorithm. In each round, the algorithm computes the marginal contribution of each element to the intermediate solution set. These oracle queries are parallelized across multiple cores. This is especially effective in settings where the oracle queries are computationally intensive;

- **LASSO.** This popular algorithm fits either a linear or logistic regression with an $\ell_1$ regularization term $\gamma$. It is known that for any given instance that is $k$-sparse there exists a regularizer $\gamma_k$ that can recover the $k$ sparse features. Using LASSO to find a fixed set of features is computationally intensive since in general, finding the regularizer is computationally intractable [MY12] and even under smoothed analysis its complexity is at least *linear* in the dimension of the problem [LS18]. We therefore used sets of values returned by LASSO

for varying choices of regularizers and use these values to benchmark the objective values returned by DASH and the other benchmarks.

## J    Observation on Worst Case Bound

In the special case of feature selection where there is no diversity term, we can get an improved approximation guarantee of $\gamma^2$, where $\gamma = m/M$.

We can bound the objective function $f(S)$ by the modular function $\sum_{a \in S} f(a)$ so that $\frac{m}{M} \sum_{a \in S} f(a) \leq f(S) \leq \frac{M}{m} \sum_{a \in S} f(a)$. Then, for the TOP-K algorithm, where we select the best $k$ elements by their value $f(a)$, we get the following approximation guarantee.

$$f(S) \geq \frac{m}{M} \sum_{a \in S} f(a) \geq \frac{m}{M} \sum_{o \in O} f(o) \geq (\frac{m}{M})^2 f(O) = \gamma^2 f(O)$$

where the first and last inequalites come from differential submodularity properties and the second inequality follows from selecting the best $k$ elements.

**Remark 22.** *In the case where $\gamma = 1$, $f(S)$ is a submodular function. In the context of feature selection, when $\gamma = 1$, the features are linearly independent and one can obtain the optimal solution by selecting the $k$ features that have the largest marginal contributions to the empty set.*