[Reviews · NeurIPS 2019]

Reviewer 1



The authors propose a relaxation of submodularity, called differential submodularity, where the marginal gains can be bounded by two submodular functions. They use this concept to provide approximation guarantees for a parallel algorithm, namely adaptive sampling, for maximizing weak submodular functions, and show its applicability to parallel feature selection and experimental design. Overall the paper is well written and the problem is well motivated. The main motivation for parallelized algorithms is their applicability to large datasets. Although we see some speedup for relatively small datasets in the experiments, my main concern is that due to the large number of rounds in the worst case and large sample complexity, the algorithm may not scale to large datasets, especially in the actual distributed setting, (e.g. MapReduce). Here are my questions: - What is the sample complexity of line 5 of Algorithm 1 (theoretically)? In experiments, the authors mention they implemented DASH with 5 samples at every round, but later they say that sampling could be computationally intensive, and that’s why DASH is slower than SDS_MA for small k. - Is the proposed algorithm scalable to large datasets, and is it applicable to actual distributed setting (e.g. MapReduce)? The number of rounds log_{1+\eps/2}(n) become prohibitive even for moderate size problems, and the experiments are done on relatively small datasets. So, the algorithm may not be actually used for large datasets that is main motivation for parallelization? - How is SDS_MA parallelized? This is not explained in the paper or appendix, and it’s wired that the parallel version is slower than the centralized one, especially in the multi-processor case and not the actual distributed setting. - Can other distributed frameworks like “A new framework for distributed submodular maximization“, "Fast greedy algorithms in mapreduce and streaming", or “Distributed submodular maximization: Identifying representative elements in massive data” be used to provide guarantees for maximizing weak submodular functions in the distributed setting (with or without differential submodularity)? ---------------------- update: I'm not very convinced by authors' answers about the fundamental difference between the current setup and MapReduce. I believe in MapReduce, the ground set can be partitioned to several machines, while each machine filters elements from its local data and samples from it, and the results can be communicated to the central machine. I also agree with other reviewers that introducing further applications could make the introduced notion of differential submodularity more interesting.

Reviewer 2



Quality/Significance: Somewhat interesting combination of previous ideas. Adaptive sampling provides good motivation for differentially submodular functions (Appendix A). Originality: The authors briefly mention [HS16] but do not explain how the definition its results A preprint by Gupta et al. is cited in the references as [GPB18] but never mentioned in the main paper or the Appendix. It contains a definition equivalent to differential submodularity in the case where g and h are the same function (as they are in many of this paper's proofs). Also, the proof of Theorem 6 is very similar to Section 5.2 of [GD18]. For these reasons, it seems like the paper combines several existing lines of work with limited novelty. Without further detailed discussion, this is a key weakness Clarity: Generally clear, but organization could be improved. Several of the paper's key details are pushed to the Appendix. d defined as both the feature dimension and a submodular diversity function The name DASH might cause confusion with ProtoDash [GDC17] [GD18] Gurumoorthy et al. Streaming Methods for Restricted Strongly Convex Functions with Applications to Prototype Selection. https://arxiv.org/pdf/1807.08091.pdf [GDC17] Gurumoorthy et al. ProtoDash: Fast Interpretable Prototype Selection. https://arxiv.org/abs/1707.01212 ---------------- EDIT: I appreciate the detailed comparison with related work provided by the authors in their response, and will increase my overall score. However there are some additional concerns about how the proposed algorithm performs in large-scale settings. Therefore my final review is still borderline

Reviewer 3



It was relatively easy to follow the paper. I like Figure 1, it helps a lot of immediately getting the definition of differential submodularity. I only do not understand why in the definition we need g_S if we already have \alpha h_S. The theoretical contribution of this work is very motivated by the prior work on performing submodular maximization in parallel, which is also noted by the authors. So, the main contribution of this paper is introducing the notion of differential submodularity. I believe that this notion has future potential, but it would be nice to see in this submission more comments and more comparison with the prior work. In general, experiments are convincing that DASH outperforms the baselines. Also, in the experiments is said :"in all experiments, DASH achieves a two to eight-fold speedup ...", and then :"This shows the incredible potential of other parallelizable algorithms". I assume that this comment refers to plots (c) and (f), i.e., the running times. I think that "incredible" here is an oversell. It is a good news that indeed one obtains faster approach, but saying "incredible" feels like it is sometime orders of magnitude more efficient. In Figure 4(a) and 4(d), when does SDS_MA saturate? It would be good to also use larger k for those experiments so ti see how SDS_MA behave. --- Review updates --- I thank the authors for providing additional comments on their work. My question still remains on the applicability of this notion to other problems (the authors mentioned that they are working on extending these results to other problems, but as I understand the work is in progress). My score remains the same.

[Author Response · NeurIPS 2019]

We thank all the reviewers for their comments and suggestions. Reviewer-specific comments to follow.

**Reviewer 1.**   Thank you for your thoughtful review. To address your main concern regarding larger scale experiments,
we ran experiments with $k = 150$ and $n = 5000$ which are larger than any of those in published works at NeurIPS and
ICML on submodularity in the past two years, with the exception of one paper using $k = 50$ and $n = 10000$. We will dis-
cuss this and explain the fundamental differences between this work on parallelization and the MapReduce framework de-
signed for distributed computing. We will appreciate if, in light of this response, you would consider revising your score.

7
• **Regarding larger scale experiments:** We ran all algorithms for $k = 150$ and
$n = 5000$ and found the results consistent with those reported in the paper (see
Figures). With the existing experimental setup described in this paper we can easily
run DASH for $k > 1000$. Note that the bottleneck is that the benchmarks such as
$\text{SDS}_{\text{MA}}$ are too slow, which is the main advantage of using DASH.

• **Regarding sample complexity of line 5 in Algorithm 1:** Please see lines 534-
539 in Appendix G. To obtain the guarantee with probability $1 - \delta$ one needs $m = $
$n(\frac{OPT}{\epsilon})^2 \log(\frac{2n}{\delta})$ samples. As discussed in lines 254-257, this is a worst-case lower
bound and, in practice, as few as 5 samples suffice. Similarly, the number of rounds
needed in practice is much lower than the theoretical number (lines 259-260).

• **Regarding applicability to MapReduce:** Yes, DASH is applicable in the MapRe-
duce setting. Algorithms in the MapReduce setting split the data across multiple
machines and run Greedy on each machine. Every such MapReduce algorithm can run
DASH instead of Greedy and enjoy a dramatic speedup. Referring to our discussion above, DASH can be implemented
on much larger instances than those that have been used in previous work, including those in the MapReduce setting.

• **Regarding parallelization of $\text{SDS}_{\text{MA}}$:** In each round of $\text{SDS}_{\text{MA}}$, the algorithm computes the marginal contri-
bution of each element to the solution set, which are parallelized. In lines 274-276, we state "When the calculation of
the marginal contribution is computationally cheap, parallelization of $\text{SDS}_{\text{MA}}$ has a longer running time...due to the
cost of merging parallelized results." We will be happy to include more details in the full version.

**Reviewer 2.**   Thank you for your review. The main concern is the lack of discussion about Theorem 6 being related to
previous work [GD18] that appears on the arXiv. There is a slight technical difference between proof of Theorem 6 and
that of [GD18]. More importantly, since this work is unpublished, we were not aware of this work at the time of writing
the paper and we will be happy to cite it. Beyond this analysis, there are many technical and conceptual contributions in
this paper that enable the exponential acceleration of statistical subset selection problems. We will appreciate if you
would consider re-evaluating your score based on our response.

• **Regarding proof of Theorem 6:** Our proof of Theorem 6 first upper bounds the marginal contribution of a single
element $a$ unlike the proof in Lemma 5.4 in GD18, which bounds the marginal contribution of the set of $A$. The
constants in the bounds also differ. Theorem 6 was introduced as an intermediate result to show that statistical subset
selection objectives are differentially submodular, which allows for effective parallelization by DASH. The definition of
differential submodularity and its application to parallelizable algorithms, which allow for both theoretical guarantees
as well as empirical performance, are the core novelties of our paper and not discussed in [GD18].

• **Regarding [HS16]:** Please see line 84 for "relaxations of submodularity and relationship to differential submodu-
larity in Appendix B" and line 439 "Horel et al. [HS16] define $\epsilon$-approximately submodular functions...". Approximate
submodularity defined in [HS16] is fundamentally different since the function is approximated pointwise by a sub-
modular function, but not its marginals. Differential submodularity stipulates that the *marginals* of a function are
approximated pointwise by submodular functions. This is a crucial difference: maximizing approximate submodular
functions leads to intractable optimization problems (for any $\epsilon \in \Omega(1/k)$ maximizing an $\epsilon$-approximate submodular
function under a cardinality constraint requires exponentially-many queries to obtain a constant factor approximation).

• **Regarding relationship to relaxation of submodularity by Gupta et al. [GPB18]:** Differential submodularity
generalizes the definition of Gupta et al. so that $g(A)$ is not equivalent to $h(A)$. This is necessary in cases where the
objective function contains a diversity factor as in lines 180-181, 187-189. Showing that functions can be lower and
upper bounded by two different functions is crucial here. We will include this in the discussion as well.

**Reviewer 3.**   Thank you for your comments. We focus on objectives that are fundamental to statistical subset selection.
We are working on extending this to dictionary selection and other applications. Regarding prior work, background
on adaptive sampling can be found in lines 37-44 and relaxations of submodularity in lines 421-441. Our differential
submodularity definition allows $g$ and $h$ to be different functions for added flexibility, which is necessary for objectives
with diversity terms (lines 180-181). Regarding experimental design, we will be happy to include results for larger $k$ to
examine $\text{SDS}_{\text{MA}}$ saturation. Regarding speedups, the bottleneck is the slowness of $\text{SDS}_{\text{MA}}$, which makes it difficult
to compare speedups for large $k$. However, we would be able to run DASH, but not $\text{SDS}_{\text{MA}}$, for $k > 1000$.

[Meta-Review · NeurIPS 2019]

First, the relation to GD18 had no bearing on decision, and none of the reviewers had a COI. This is a borderline paper. The reviewers liked the notion of differential submodularity, and the avenues of future work it may open. That said, there are several things that need to be improved before the paper is ready for publication: 1. take all reviewer comments into account 2. add truly large scale experiments that really test the new method and compare it to existing ones. Only this will show the true performance. Please see the scale of experiments e.g. in these papers: http://proceedings.mlr.press/v48/mirzasoleiman16.html http://proceedings.mlr.press/v70/mirzasoleiman17a.html http://proceedings.mlr.press/v80/mitrovic18a.html http://proceedings.mlr.press/v97/kazemi19a.html These types of experiments are state of the art and need to be included. 5000 is too small. 3. Add more examples and applications. To cite from the discussion: "My main concern/question is the applicability to other problems. That is, with extra applications to other problems it would be more convincing that this notion is interesting for further investigation." These improvements are a 'must'.